# Modelling the Sintering of Nickel Particles Supported on γ-Alumina under Hydrothermal Conditions

**Isabelle Champon** [1,2,3], **Alain Bengaouer** [1], **Albin Chaise** [1], **Sébastien Thomas** [2] and **Anne-Cécile Roger** [2,*] 

1  CEA, LITEN, DTBH, SCTR, LER, Université Grenoble Alpes, 17 Avenue des Martyrs,
   F-38054 Grenoble, France; isabelle.champon@gmail.com (I.C.); alain.bengaouer@cea.fr (A.B.);
   albin.chaise@cea.fr (A.C.)
2  ICPEES-ECPM, UMR 7515, University of Strasbourg, 25 rue Becquerel, F-67087 Strasbourg, France;
   sebastien.thomas@unistra.fr
3  French Environment and Energy Management Agency, 20 Avenue du Grésillé BP 90406,
   49004 Angers, France
*  Correspondence: annececile.roger@unistra.fr

**Abstract:** Sintering of nickel particles is a well-known path of deactivation for $Ni/Al_2O_3$ catalysts. Considering the $CO_2$ methanation in the context of Power-to-Gas, a sintering study for up to 300 h was performed in a controlled atmosphere between 450 and 600 °C. Since water is a product of the methanation reaction and is known to favor the particle sintering, the $H_2O:H_2$ molar ratio was varied in the range 0–3.2. Characterization of the post mortem samples showed sintering of both nickel and support particles. The absence of carbon oxides in the gas feed allows us to rule out other causes of deactivation such as carbon deposits. A sintering law is derived from the loss of metallic surface area with time-on-stream according to local temperature and $H_2O:H_2$ molar ratio. An excellent fit of the experimental data was obtained allowing the prediction of the metallic surface area within 15%.

**Keywords:** carbon dioxide; methanation; Power-to-Gas; deactivation; sintering; $Ni/Al_2O_3$ catalyst

## 1. Introduction

Power-to-SNG (Substitute Natural Gas) aims at storing and transporting the surplus production of renewable energy [1–6] as SNG in the infrastructures already existing for the natural gas. Hydrogen is generated by water electrolysis using renewable electricity and combined with carbon dioxide through the $CO_2$ methanation; also called the Sabatier reaction. When the $CO_2$ is biogenic, this process leads to renewable SNG with no net $CO_2$ emission.

The $CO_2$ reduction into methane in the gas phase is a difficult step since carbon dioxide is a very stable molecule. Thus, a catalyst is needed in order to obtain acceptable kinetics and methane selectivity. A supported catalyst is generally used for the $CO_2$ methanation reaction and consists of a metallic phase (Rh, Ru, Ni . . . ) dispersed on an oxide support with a large surface area ($Al_2O_3$, $CeO_2$, $SiO_2$ . . . ).

Regardless of the reactor, catalyst deactivation happens with time-on-stream and is characterized by a loss of catalyst activity and/or selectivity which, at the scale of a Power-to-Gas unit, can lead to a degradation of the quality of the SNG produced, until the SNG no longer meets the gas network specifications.

Bartholomew [7] proposed in 2001 a classification of the different paths of deactivation for a supported catalyst. Three types of deactivation appeared: chemical, thermal and mechanical. The first one considered poisoning, formation of volatile compounds and gas-solid or solid-solid reactions,

and the second took account of the possible thermal degradation of the active phase or the support by sintering. Mechanical deactivation was for fouling and attrition/crushing of the grains.

In the work presented here, a milli-structured fixed-bed reactor exchanger is used [8,9]. The main goal is for the SNG produced to directly meet the gas network specifications and consequently to limit the different stages of gas purification. Indeed, very high conversion rates are obtained in these reactors since kinetics are very fast thanks to the exothermicity of the methanation reaction. In these reactors, temperature up to 600 °C can be reached locally.

In order to predict the performances of the milli-structured fixed-bed reactor-exchanger with time-on-stream, the understanding and modeling of catalyst deactivation kinetics is needed. The modeling of initial kinetics on the same catalyst has already been published [10]. In the literature, several studies dealt with nickel particle sintering in the conditions of $CO_2$ methanation studied here, assuming that sintering is the principal cause of catalyst deactivation.

In 2007, Rostrup-Nielsen et al. [11] dealt with methanation of synthesis gas from coal. An adiabatic reactor was used with a recycle of the product gas for more than 8000 h. At 600 °C, the MCR2X catalyst, constituted of 22 wt % Ni on a stabilized support, showed a loss of active surface area. They successfully fitted the relative nickel particles diameters with time on stream as a simple power law, using $k_{sint}$ and $n$ as parameters:

$$\frac{d_{Ni}(t)}{d_{Ni°}} = (1 + k_{sint}t)^n \tag{1}$$

They concluded both a loss of active surface area and a loss of activity. Furthermore, they found a loss of activity per unit surface area (turn-over frequency).

According to Ruckenstein and Pulvermacher [12], a simple power law described the exposed metallic surface area with time:

$$-\frac{d\left(\frac{D(t)}{D_0}\right)}{dt} = k_{sint}\left(\frac{D(t)}{D_0}\right)^n \tag{2}$$

$D_0$, $k_{sint}$ and $n$ being initial dispersion, sintering kinetic constant and sintering order, respectively.

A generalized power law was then proposed by Fuentes [13], who considered the asymptotic limit of dispersion at infinite time $D_{eq}$:

$$-\frac{d\left(\frac{D(t)}{D_0}\right)}{dt} = k_{sint}\left(\frac{D(t)}{D_0} - \frac{D_{eq}}{D_0}\right)^n \tag{3}$$

Sehested et al. [14,15] studied the sintering of nickel-based catalysts used in a steam reforming process. They concluded that not only time and temperature influenced the nickel particle sintering but also the $H_2O:H_2$ ratio.

In 2019, Ewald et al. [16] studied the deactivation of several as-prepared Ni catalysts used in $CO_2$ methanation. Catalysts were aged at 250, 300 and 350 °C under equilibrium conditions up to 165 h. Severe deactivation was revealed and a power law model [17] was applied for the description of the deactivation kinetics:

$$-\frac{da}{dt} = k_d a^d \tag{4}$$

$a$, $k_d$ and $d$ being activity, deactivation kinetic constant and reaction order of deactivation, respectively.

Detailed characterization of post mortem samples allowed us to establish the main causes of deactivation. They were found to be Ni particle sintering associated with a global loss of BET (Brunauer, Emmett and Teller) surface, reduction of $CO_2$ adsorption capacity and especially number of medium basic sites and structural changes of the mixed oxide phase.

Assuming nickel particle sintering is the principal cause of catalyst deactivation [11,14,15] the aim of the present study is to develop a sintering law in order to model the loss of catalyst activity with time on stream. To do so, aging tests were carried out in a controlled atmosphere and in the

absence of carbon oxides in order to rule out other causes of deactivation, such as carbon deposits. The objectives of these tests were to quantify, on one hand, the influence of temperature, and on the other hand, the influence of the reactive atmosphere ($H_2O$:$H_2$ molar ratio since water is a product of the methanation reaction) on the metallic structure and surface. The time evolution of metallic surface is then used to identify the different parameters of a General-Power-Law published in the literature.

## 2. Results and Discussion

Reducibility of the catalyst was studied by $H_2$-TPR (Hydrogen-temperature programmed reduction). TPR profile (Figure 1a) showed two zones: the first one, around 330 °C, was associated with the reduction of bulk NiO and the second one, around 520 °C, was associated with NiO in strong interaction with $Al_2O_3$. At this temperature, it could be the reduction of a $NiAl_2O_4$ spinel. The activation procedure at 300 °C allowed to reduce 43% of the Ni present in the sample (Figure 1b).

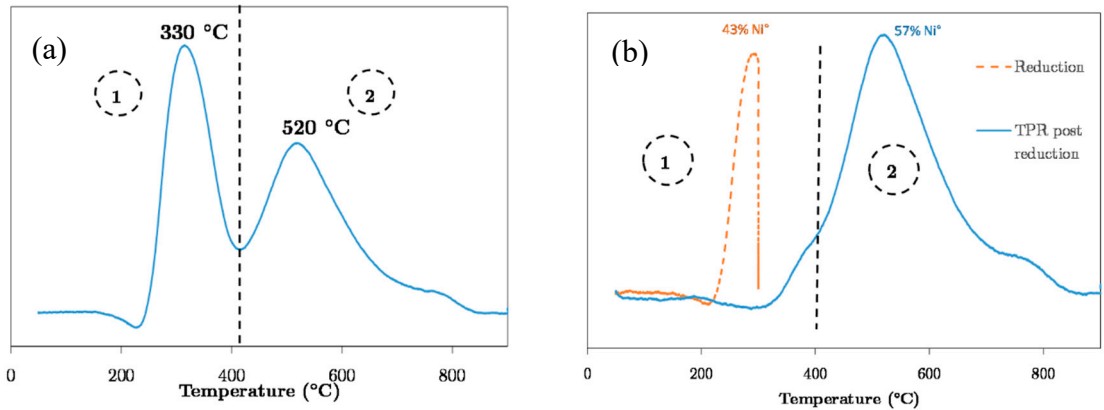

**Figure 1.** (**a**) $H_2$-TPR of the fresh commercial catalyst Ni/$Al_2O_3$; (**b**) Activation procedure at 300 °C followed by a $H_2$-TPR from 50 until 900 °C.

$CO_2$ methanation catalyst activity measurements were performed on fresh and post mortem catalysts aged for different times in hydrothermal conditions ($H_2O$/$H_2$ = 0.26; T = 600 °C). Catalytic tests were conducted at 350 °C with 10 mg of catalyst with a total flow of 55 NmL min$^{-1}$ with molar ratio: $H_2$/$CO_2$/$N_2$ = 73/18/9. $CO_2$ conversions and $CH_4$ yields are presented Figure 2a. As expected, both decrease with increasing hydrothermal ageing time. The loss of activity can be clearly linked to the decrease of metallic surface (Figure 2b). This validates the fact that Ni sintering is a major cause of the catalyst deactivation.

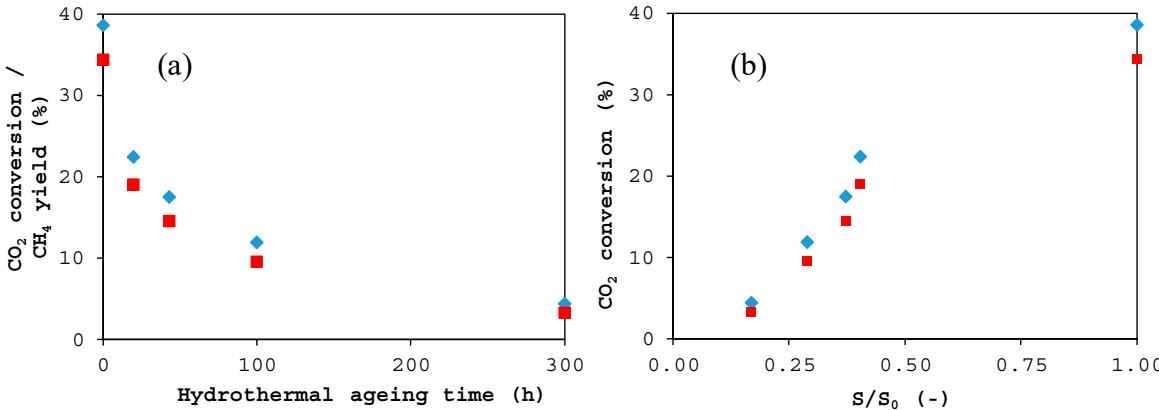

**Figure 2.** Catalyst activity measurements at 350 °C of fresh and post mortem samples aged at 600 °C in hydrothermal conditions ($H_2$:$H_2O$ = 0.26) as a function of hydrothermal ageing time (**a**) and relative metallic surface (**b**) ◆: $CO_2$ conversion ■: $CH_4$ yield.

Loss of metallic surface due to metal particles sintering was quantified by metal surfaces estimation with hydrogen chemisorption and programmed temperature desorption (TPD-$H_2$) on post mortem samples. Metallic surfaces were calculated according to the quantity of adsorbed (or desorbed) hydrogen $N_{H_2}$ (mol·g$^{-1}$):

$$S_{Ni°} = n\, N_{H_2}\, N_A\, a_m \qquad \left(m^2\, g^{-1}\right) \tag{5}$$

$n$ is the stoichiometry of the chemisorption reaction, equal to 2 in the case studied here of a dissociative chemisorption of $H_2$ on Ni° [18], $N_A$ is the Avogadro number, $a_m$ corresponds to the average surface area occupied by a Ni atom in a face-centered cubic structure for an equivalent exposure of the (111), (100) and (110) planes, i.e., $6.51 \times 10^{-20}$ m$^2$ [19].

An average particle size can be deduced by assuming that these particles are half-spheres on the surface of the support:

$$d = \frac{6\, wt.\left(\frac{m}{\rho}\right)}{S} \tag{6}$$

*wt.* is the metallic weight percentage present in the sample after reduction and is equal to 5.2%. *m* is the mass of the reduced sample.

Figure 3 shows the evolution of the metallic surface estimated by $H_2$ chemisorption with time-on-stream for the different conditions of temperature and atmosphere tested. No evolution of the metallic surface was observed in 200 h of thermal aging i.e., hydrogen atmosphere at 450 °C (Figure 3a). The Ni/Al$_2$O$_3$ commercial catalyst was, therefore, thermally stable under these conditions (H$_2$:N$_2$). Similarly, the metallic surface was stable for 100 h of thermal aging at 600 °C. However, a loss of metal surface was observed between 100 and 200 h of thermal aging at 600 °C. In conclusion, the nickel metallic surfaces obtained for 20 h of thermal aging at 450 and 600 °C were used as initial values of surface for the hydrothermal samples aged at 450 and 600 °C, and both samples aged 20 h in thermal conditions (H$_2$:N$_2$) were used as reference samples for the hydrothermal aging.

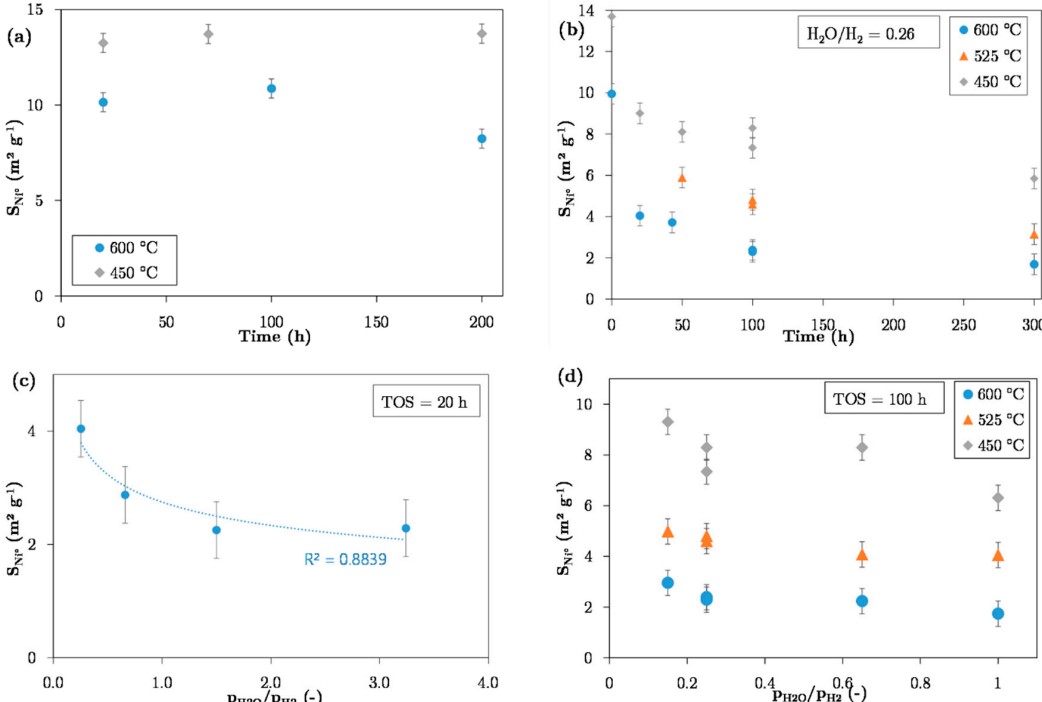

**Figure 3.** (**a**) Effect of temperature on metal particle sintering in reductive atmosphere H$_2$:N$_2$; (**b**) effect of temperature on metal particle sintering under H$_2$:H$_2$O atmosphere, with a H$_2$O to H$_2$ ratio of 0.26; effect of H$_2$O to H$_2$ ratio on metal particle sintering (**c**) for 20 h at 600 °C and (**d**) for 100 h at 450, 525 and 600 °C.

Whatever the temperature, the nickel surface area loss in hydrothermal conditions was very fast between 0 and 20 h and moderate between 20 and 300 h (Figure 3b). In addition, the effect of temperature on the metal particles sintering was highlighted. Indeed, the loss of metal surface was 60% for 20 h of aging at 600 °C against 40% at 450 °C.

A power law seems to be able to describe the evolution of the metallic surface with the $H_2O:H_2$ ratio for a given time on stream (Figure 3c). Moreover, the effect of aging temperature on metal particle sintering is greater than the effect of $H_2O:H_2$ molar ratio in the studied range (Figure 3d).

Specific surface area of the post mortem samples was measured by nitrogen physisorption. Evolutions of the specific surface area with time are a proof of support sintering in hydrothermal aging (Figure 4). Indeed, a loss of 38% of the specific surface area was observed for 20 h of hydrothermal aging at 600 °C against 12% at 450 °C with a $H_2O:H_2$ molar ratio of 0.26; as temperature favors the support sintering. Moreover, in thermal aging (without water) at 600 °C, no loss of specific surface area was observed for 20 h of aging, showing that water has a real impact on the sintering of the support.

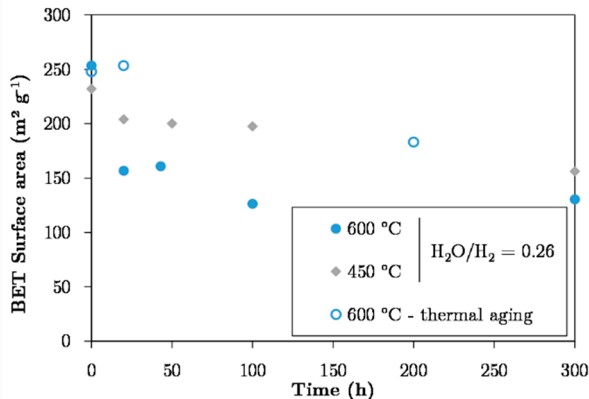

**Figure 4.** Evolution of the specific surface area with time at 450 °C and 600 °C in thermal and hydrothermal aging.

The crystalline phases constituting the samples were determined from the diffractograms presented in Figure 5. The crystalline phases identified are the metallic nickel cfc (JCPDS 03-065-2865) by the three diffraction lines corresponding to the crystallographic planes (111), (200) and (220); the γ-alumina cfc (JCPDS 00-010-0425) by the diffraction lines corresponding to the crystallographic planes (400) and (440) and the spinel $NiAl_2O_4$ cfc (JCPDS 00-010-0339), by the diffraction line of the crystallographic plane (400) which overlaps on that of alumina.

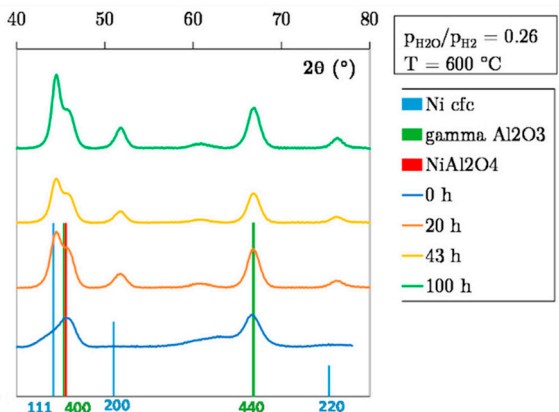

**Figure 5.** XRD pattern of the commercial $Ni/Al_2O_3$ catalyst aged in hydrothermal conditions at 600 °C with a $H_2O$ to $H_2$ molar ratio of 0.26.

The emergence of the nickel diffraction lines is well visible with time-on-stream (Figure 5) which is an evidence of nickel sintering. On the reference sample the nickel diffraction lines are hardly discernible, probably because nickel crystallites were very small (<5 nm). However, for the sample aged 20 h, the nickel diffraction lines distinguish themselves. Nevertheless, little variation is observed on the diffraction profile between 20 and 100 h. In addition, the values of the nickel crystallites diameter calculated by the Debye-Scherrer formula are close to the validity limit of the formula (5 nm). It is likely that no significant change in crystallite size occurred between 20 and 100 h aging at 600 °C.

Transmission electron microscopy (TEM) images (Figure 6) allowed to establish a nickel particle size distribution for the catalyst samples aged for 20, and 100 h in hydrothermal conditions ($H_2O:H_2$ = 0.26). For each TEM photography, the averaged metallic particle size is obtained from different measurements in two orthogonal directions. A sample of about thirty particles is counted to obtain a size distribution. It was found relatively monodispersed distribution with a growth of the number average metallic particle size from 6.5 nm for the sample aged 20 h to 10.5 nm for the sample aged 100 h. Number average particle sizes estimated by TEM are close to surface average particle sizes obtained by chemisorption assuming the particles to be half-spheres (Table 1), the number average being logically lower than the surface average. This is indicative of a decrease of metallic surface by Ni sintering rather that by encapsulation due to support sintering.

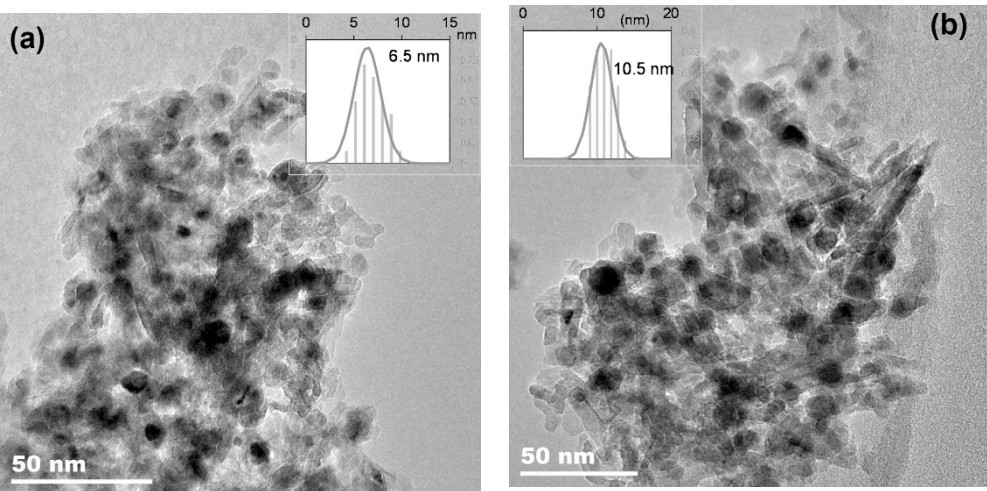

**Figure 6.** TEM images and nickel particle size distribution for the catalyst samples aged for 20 h (**a**) and 100 h (**b**) in hydrothermal conditions at 600 °C with a $H_2O$ to $H_2$ ratio of 0.26.

**Table 1.** Comparison of the average particle sizes estimated from TEM images and $H_2$ chemisorption for samples aged in hydrothermal conditions ($H_2O:H_2$ = 0.26) at 600 °C.

| Time-on-Stream (h) | TEM Average Diameter (nm) | Chemisorption Average Diameter (nm) Calculated by Equation (6) |
|:---:|:---:|:---:|
| 20 | 6.5 | 8.6 |
| 100 | 10.5 | 12.3 |

The Generalized-Power-Law-Expression (GPLE) proposed by Fuentes [13] was adapted in order to take into account the aging atmosphere by considering that the $H_2O:H_2$ ratio promotes the sintering.

$$-\frac{d\left(\frac{S(t)}{S_0}\right)}{dt} = k_{sint,ref}\exp\left(-\frac{Ea_{sint}}{R}\left(\frac{1}{T_{ref}} - \frac{1}{T}\right)\right)\left(\frac{S(t)}{S_0} - \frac{S_{eq}}{S_0}\right)^2\left(\frac{p_{H_2O}}{P_{H_2}}\right)^p \qquad (7)$$

The four parameters of the sintering law were the sintering kinetic constant at the reference temperature of 600 °C ($k_{sint,ref}$), the sintering activation energy ($Ea_{sint}$), the equilibrium metallic surface i.e., metallic surface reached at infinite sintering time ($S_{eq}$) and the exponent $p$.

The equation was numerically integrated and the four parameters were identified by the method of least squares minimization.

For an initial metallic surface of 10 m$^2$ g$^{-1}$, and a reference temperature of 600 °C, the identified parameters are given in Table 2.

**Table 2.** Identified parameters of the sintering law for an initial metallic surface area of 10 m$^2$ g$^{-1}$ and a reference temperature of 600 °C.

| $p$ | $k_{sint,ref}$ (h$^{-1}$) | $Ea_{sint}$ (kJ mol$^{-1}$) | $S_{eq}/S_0$ |
| --- | --- | --- | --- |
| 0.63 | 0.32 | 126 | 0.19 |

A comparison between the experimental and modeled loss of metallic surface area is given for Figure 7 300 h hydrothermal aging with a H$_2$O:H$_2$ molar ratio of 0.26 at 450, 525 and 600 °C (a), as well as for several H$_2$O:H$_2$ ratios for 100 h hydrothermal aging at 450, 525 and 600 °C (b), and for 20 h hydrothermal aging at 600 °C (c). The model predicts with an accuracy of 15% the metallic surface area for 300 h aging between 450 and 600 °C for a H$_2$O:H$_2$ ratio ranging from 0 to 3.2 (Figure 8).

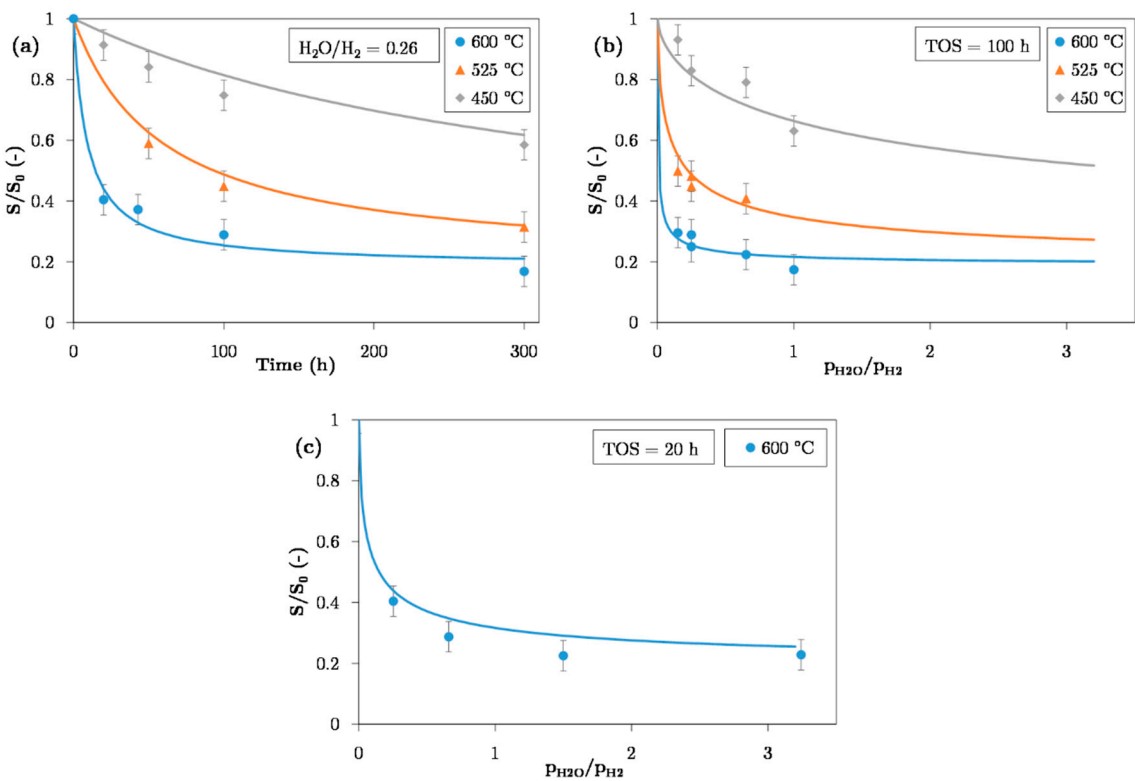

**Figure 7.** Comparison of experimental and modeled loss of metallic surface area (**a**) for 300 h hydrothermal aging at 450, 525 and 600 °C under H$_2$:H$_2$O atmosphere, with a H$_2$O to H$_2$ ratio of 0.26; for several H$_2$O to H$_2$ ratios (**b**) for 100 h at 450, 525 and 600 °C and (**c**) for 20 h at 600 °C.

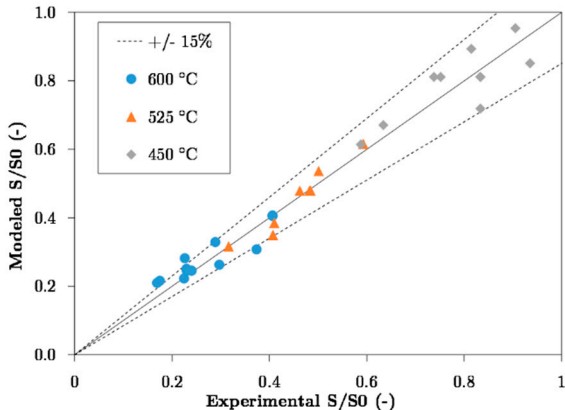

**Figure 8.** Parity plot of the experimental and modeled metallic surface area up to 300 h aging within 15% error at 450, 525 and 600 °C for a $H_2O:H_2$ ratio ranging from 0 to 3.2.

## 3. Experimental Study

### 3.1. Equipment and Materials

Aging tests were performed at atmospheric pressure, between 450 and 600 °C on a 14–17 wt.% $Ni/Al_2O_3$ commercial catalyst (mass of around 0.50 g).

The setup consisted of hydrogen and nitrogen supply regulated by mass flowmeters (MFC). The nitrogen flow was used to conserve a constant total flow rate. Liquid water was injected using a micro pump (GILSON 321) and mixed with the gases in a vaporization chamber. All gas lines were heated after the vaporization chamber to avoid water condensation. Water was condensed and separated from the gases downstream the aging zone.

The aging zone consisted of a tubular furnace for heating the quartz tubular reactor to the desired temperature. The reactor was 30 cm long and $^1/_4$-inch inner diameter. The catalyst bed (height = 1.5 cm), consisting of catalyst particles, in the form of powder, sieved between 300 and 400 μm, was maintained by quartz wool and located in the isothermal zone of the oven.

Three thermocouples were used in this process: a first thermocouple controlled the temperature of the furnace, a second thermocouple measured the temperature inside the reactor (at 0.5 cm below the catalytic bed), and a third thermocouple controlled the temperature of the vaporization chamber (200 °C).

Prior to aging tests, the fresh catalyst samples were activated in situ at 5 °C min$^{-1}$ until 300 °C followed by 300 min at 300 °C with a 10% $H_2$/Ar mixture (42 NmL min$^{-1}$).

Then, the effect of temperature on metal particle sintering was studied in reductive atmosphere. Several samples were aged for 20, 100 and 200 h at 450 and 600 °C with 50 NmL min$^{-1}$ of hydrogen and 25 NmL min$^{-1}$ of nitrogen. This study is called in the following the "thermal aging" and was realized to establish the effect of temperature only (without water) on the metallic surface with time on stream.

Thereafter, the effect of temperature on metal particle sintering was studied under $H_2:H_2O$ atmosphere, with a $H_2O:H_2$ molar ratio of 0.26. Several samples were aged for 20, 50, 100 and 300 h at 450, 525 and 600 °C with 50 NmL min$^{-1}$ of hydrogen, 25 NmL min$^{-1}$ of nitrogen and 10 μL min$^{-1}$ of liquid water meaning 13 NmL min$^{-1}$ of steam. This study is called the "hydrothermal aging" in the following and is realized to establish the effect of temperature on the metallic surface with time on stream for a given $H_2O:H_2$ molar ratio.

The second part of the "hydrothermal aging" study is to consider the effect of atmosphere composition on metal particle sintering under $H_2:H_2O$ atmosphere with a total flow rate of 88 NmL min$^{-1}$.

For 20 h at 600 °C, several $H_2O:H_2$ molar ratios were studied: 0.26, 0.70, 1.5, and 3.2.

For 100 h at 450, 525, and 600 °C, several $H_2O:H_2$ molar ratios were studied: 0.15, 0.26, 0.70, and 1.0.

Used catalyst samples are termed "aged" or "post mortem" and opposed to "fresh" catalyst samples, i.e., not used.

*3.2. Catalyst Characterization*

Hydrogen temperature programmed reduction ($H_2$-TPR) was carried out with a Micromeritics Autochem II equipped with a thermal conductivity detector. In total, 0.10 g of the catalyst was used for the analysis. Pretreatment of the samples at 10 °C min$^{-1}$ until 200 °C and 1 h at 200 °C under Ar (50 NmL min$^{-1}$), was followed by cooling until 50 °C. The analysis was then started in a 10% $H_2$/Ar mixture (50 NmL min$^{-1}$) at 5 °C min$^{-1}$ until 900 °C.

Metallic surface area was determined by hydrogen-pulsed chemisorption and subsequent hydrogen temperature programmed desorption ($H_2$-TPD) using a Autochem II (Micromeretics, Norcross, GA, USA). First, the post mortem samples were activated at 5 °C min$^{-1}$ until 300 °C and 30 min at 300 °C with a 10% $H_2$/Ar mixture (50 NmL min$^{-1}$) since post mortem samples were re-oxidized at ambient atmosphere. Then, the samples were cooled down until 50 °C and 10% $H_2$ in Ar were pulsed onto the samples every 4.5 min. In total, 45 pulses were injected with a 500 μL loop (maintained at 110 °C and atmospheric pressure), equivalent to 1.59 μmol of hydrogen. Subsequently, the samples were flushed and heated up under argon (50 NmL min$^{-1}$) until 900 °C at 15 °C min$^{-1}$.

The specific surface area of the catalysts was estimated by $N_2$ physisorption at 77 K. Prior to the analysis, the samples were degassed at 200 °C during one night under vacuum. The BET method was used to determine the specific surface area using a ASAP 2420 ((Micromeretics, Norcross, GA, USA).

X-ray diffraction (XRD) measurements of the samples were conducted with a D8 Advance (Bruker, Billerica, MA, USA) equipped with a copper source and with a Lynxeye photodiode detector. Measurements were carried out within the range 40–80°. Average sizes of the Ni° crystallites were estimated with the Scherrer equation from the (200) crystallographic plane of nickel metal.

Transmission electron microscopy snapshots were taken on a Jeol JEM-2100 microscope (Jeol, Tokyo, Japan) in order to establish a particle size distribution.

# 4. Conclusions

Characterization of the post mortem samples allowed us to characterize both particle and support sintering. Assuming half spherical particles, TEM images validated the estimation of the average particle diameter by hydrogen chemisorption.

A Generalized Power-Law Expression (GPLE) taking into account temperature and local atmosphere was identified in order to model the nickel particle sintering of a commercial catalyst (Ni/Al$_2$O$_3$) with time-on-stream at temperatures between 450 and 600 °C, and H$_2$O:H$_2$ ratio between 0 and 3.2.

$$-\frac{d\left(\frac{S(t)}{S_0}\right)}{dt} = 0.32\ \exp\left(-1.52\ 10^4 \left(\frac{1}{873} - \frac{1}{T\ [K]}\right)\right)\left(\frac{S(t)}{S_0} - 0.19\right)^2 \left(\frac{p_{H_2O}}{P_{H_2}}\right)^{0.63} \tag{8}$$

For 300 h on-stream, the model allows to predict the metallic surface area within 15% in the conditions previously cited.

Future works will include the evolution of the catalyst activity with time-on-stream in CO$_2$ methanation reactive conditions (H$_2$/CO$_2$ = 4.0), and the comparison of the experimental deactivation with modeled one through the coupled model of initial CO$_2$ methanation kinetics [10] with the sintering law presented here.

**Author Contributions:** Data curation, I.C. and A.C.; Methodology, A.B. and S.T.; Supervision, A.B., A.C., S.T. and A.-C.R.; Validation, A.-C.R.; Writing—original draft, I.C.; Writing—review & editing, A.-C.R. All authors have read and agreed to the published version of the manuscript.

**Funding:** This work was supported by the French Environment and Energy Management Agency (ADEME), French Alternative Energies and Atomic Energy Commission (CEA) and ATMOSTAT. It is the result of experiments conducted at the ICPEES Laboratory in Strasbourg.

**Conflicts of Interest:** The authors declare no conflict of interest.

## Nomenclature

| | |
|---|---|
| *a* | Activity |
| $a_m$ | Average surface area occupied by a Ni atom in a face-centered cubic structure |
| *d* | Particle diameter |
| *D* | Metallic dispersion |
| *Ea* | Activation energy (J mol$^{-1}$) |
| $k_d$ | Deactivation constant |
| $k_{sint}$ | Sintering constant (h$^{-1}$) |
| $N_A$ | Avogadro number |
| $N_{H_2}$ | Quantity of adsorbed (or desorbed) $H_2$ (mol g$^{-1}$) |
| $p_i$ | Partial pressure of component *i* (Pa) |
| *R* | Ideal gas constant (R = 8.314 J mol$^{-1}$ K$^{-1}$) |
| *S* | Metallic surface (m$^2$ g$^{-1}$) |
| t | Time (h) |
| *T* | Temperature (K) |
| *wt*. | metallic weight percentage (%) |

## Greek Letters

| | |
|---|---|
| $\rho$ | Nickel bulk density (kg m$^{-3}$) |

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
