# Peer review of "Modelling the Sintering of Nickel Particles Supported on γ-Alumina under Hydrothermal Conditions"

_catalysts, doi:10.3390/catal10121477_

Round 1

Reviewer 1 Report

The material presented in the document is properly represented by numerous analyzes such as BET, XRD, TEM with particle size distribution.

The proposed methodology that has been described is also not controversial.

If we relate the content presented by the authors to the title only, it can be considered a finished work. However, I feel unsatisfied due to the lack of activity results for the described catalyst samples in the methanation reaction.

I understand that the tested catalyst system (nickel on alumina) is a commercial catalyst. So why do we know so little about its activity under working conditions? The authors themselves mention in the text that one of the deactivation possibilities is the formation of a carbon deposit. It is also possible to poison the catalyst with impurities that may be contained in the industrial gas fed to the reaction.

For me, a parameter that has not been correlated with the size of nickel particles, BET and XRD results is catalytic activity, which in fact is the essence of the considerations in the text.

In my opinion, it would be beneficial for the publication to supplement the presented results with the activity of the catalyst in the model system (e.g. CO + H2 or CO2 + H2) for post mortem samples, which the authors probably have archived.

I believe that including these results, compared to the number of analyzes included in the text, does not involve a great deal of effort.

Author Response

-Reviewer 1

The material presented in the document is properly represented by numerous analyzes such as BET, XRD, TEM with particle size distribution.

The proposed methodology that has been described is also not controversial.

Responses:The authors are indebted to the reviewer for his deep analysis of the present manuscript. Special attention was paid to his comments and a response is provided in each case as well as all modifications in the manuscript if needed.

If we relate the content presented by the authors to the title only, it can be considered a finished work. However, I feel unsatisfied due to the lack of activity results for the described catalyst samples in the methanation reaction.

Responses:Activity in CO2 methanation at 350°C for different ageing times has been added in Figure 2 a). These activities are also plotted as a function of the metallic surface (Figure 2 b). Text has been modified accordingly, by the addition of lines 97-103.

“CO2 methanation catalyst activity measurements were performed on fresh and post mortem catalysts aged for different times in hydrothermal conditions (H2O/H2 = 0.26; T= 600°C). Catalytic tests were conducted at 350 °C with 10 mg of catalyst with a total flow of 55 Nml.min-1 with molar ratio : H2/CO2/N2 = 73/18/9. CO2 conversions and CH4 yields are presented Figure 2 (a). As expected, both CO2 conversion and methane yield decrease with increasing hydrothermal ageing time. The loss of activity can be clearly linked to the decrease of metallic surface (Figure 2 (b)). This validates the fact that Ni sintering is a major cause of the catalyst deactivation.”

Figure 2. Catalyst activity measurements at 350 °C of fresh and post mortem samples aged at 600 °C in hydrothermal conditions (H2:H2O = 0.26) as a function of hydrothermal ageing time (a) and relative metallic surface (b) ¨: CO2 conversion ■ CH4 yield

I understand that the tested catalyst system (nickel on alumina) is a commercial catalyst. So why do we know so little about its activity under working conditions? The authors themselves mention in the text that one of the deactivation possibilities is the formation of a carbon deposit. It is also possible to poison the catalyst with impurities that may be contained in the industrial gas fed to the reaction.

Responses:The commercial catalyst Ni/Al2O3 is used for CO2 methanation in this study. Not so much articles deal with CO2 methanation and describe the deactivation of this catalyst in literature.

As previously mentioned, activities of the fresh and post mortem catalyst have been added Figure 2 a) and correlation of these activities with metallic surface added Figure 2 b) respectively with addition in the text that deactivation is strongly linked to nickel sintering.

For me, a parameter that has not been correlated with the size of nickel particles, BET and XRD results is catalytic activity, which in fact is the essence of the considerations in the text.

In my opinion, it would be beneficial for the publication to supplement the presented results with the activity of the catalyst in the model system (e.g. CO + H2 or CO2 + H2) for post mortem samples, which the authors probably have archived.

I believe that including these results, compared to the number of analyzes included in the text, does not involve a great deal of effort.

Responses:Results of CO2 methanation catalytic activity of the post mortem catalysts have been added (cf Figure 2 and line 97-103.

Reviewer 2 Report

The authors describe a study about "modelling the sintering of nickel particles supported on γ-alumina under hydrothermal conditions". The subject - deactivation of Ni/Al2O3 catalysts for the CO2 methanation - is of high interest, the experiments are well chosen. Some minor points should be clarified or explained before publishing:

Figure 1: Please explain the number in the circles and line at 400°C. How do you determine, that 43 % of the Ni was reduced after the activation.

Figure 2: Please explain how do you determine the uncertainties of the measurements. The points at 0 h are hardly visible.

Figure 4: There are clear changes of the diffraction pattern between 40° and 50°. Please explain it. Are there any changes of NiAl2O4?

Figure 5: How many particles do you use for the particle size distribution? How do determine the distribution? Do you use any software packages.

A general question: Is this a sintering or a particle growth? The authors should give some comment on this question.

The authors gave an exciting outlook to compare the results of sintering with the deactivation, but I understand that they want to publish it in a seperate manuscript.

Author Response

-Reviewer 2

The authors describe a study about "modelling the sintering of nickel particles supported on γ-alumina under hydrothermal conditions". The subject - deactivation of Ni/Al2O3 catalysts for the CO2 methanation - is of high interest, the experiments are well chosen. Some minor points should be clarified or explained before publishing:

Responses: The authors are indebted to the reviewer for his deep analysis of the present manuscript. Special attention was paid to his comments and a response is provided in each case as well as all modifications in the manuscript if needed.

Figure 1: Please explain the number in the circles and line at 400°C. How do you determine, that 43 % of the Ni was reduced after the activation.

Responses:The numbers in circles are used to refer to the two zones of reduction: the first one is the reduction of NiO and the second one is the reduction of the NiAl2O4. The line at 400 °C allows to separate the two zones of reduction.

The value 43% corresponds to the proportion of H2 consumed in the low temperature range with respect to the total consumption. The sentence “reduce 43 % of the nickel present” is based on the assumption that nickel is the only reducible cation and that it is fully reduced to Ni metal after the TPR experiment.

Figure 2: Please explain how do you determine the uncertainties of the measurements. The points at 0 h are hardly visible.

Responses:The uncertainties on the surface measurements are linked to the TPD analysis precision measurement. Previous studies made on this apparatus led to determine that the precision of hydrogen amount measurement was around 0.6 µmolH2. Taking into account the constant mass of sample used for TPD analysis (150 mg) and the exposed surface of a nickel atom, the uncertainities of the surface measurements are +/- 0.3 m².g-1.

Figure 4: There are clear changes of the diffraction pattern between 40° and 50°. Please explain it. Are there any changes of NiAl2O4?

Responses:As discussed in lines 154-155, the changes in the pattern at 40-50° are attributed to the growth of Ni particles, since the diffraction lines corresponding to the nickel cfc at 52° and 77 ° also much sharper than at t = 0h. Minor changes of the support diffraction lines are also visible is this range, due to the sintering of the support  evidenced with BET analysis.

Figure 5: How many particles do you use for the particle size distribution? How do determine the distribution? Do you use any software packages.

Responses:A sample of about thirty particles is counted to obtain a size distribution. Details have been added lines 166-168.

“TEM images allowed to establish a nickel particle size distribution for the catalyst samples aged for 20, and 100 h in hydrothermal conditions (H2O:H2 = 0.26). For each TEM photography, the averaged metallic particle size is obtained from different measurements in two orthogonal directions. A sample of about thirty particles is counted to obtain a size distribution. It was found relatively monodispersed distribution with a growth of the number average metallic particle size from 6.5 nm for the sample aged 20 h to 10.5 nm for the sample aged 100 h. Number average particle sizes estimated by TEM are close to surface average particle sizes obtained by chemisorption assuming the particles to be half-spheres (Table 1), the number average being logically lower than the surface average.”

A general question: Is this a sintering or a particle growth? The authors should give some comment on this question.

Responses:The growth of Ni particle has been highlighted by TEM photography, XRD and H2 chemisorption. A sentence has been added in lines 173-174.

“This is indicative of a decrease of metallic surface by Ni sintering rather that by encapsulation due to support sintering.”

The authors gave an exciting outlook to compare the results of sintering with the deactivation, but I understand that they want to publish it in a seperate manuscript.

Responses:These results will indeed be included and discussed in another manuscript.
